# Workaholism, Intensive Smartphone Use, and the Sleep-Wake Cycle: A Multiple Mediation Analysis

**DOI:** 10.3390/ijerph16193517

**Published:** 2019-09-20

**Authors:** Paola Spagnoli, Cristian Balducci, Marco Fabbri, Danila Molinaro, Giuseppe Barbato

**Affiliations:** 1Department of Psychology, University of Campania “Luigi Vanvitelli”, CE 81100 Caserta, Italy; paola.spagnoli@unicampania.it (P.S.); Marco.fabbri@unicampania.it (M.F.); danila.molinaro@studenti.unicampania.it (D.M.); giuseppe.barbato@unicampania.it (G.B.); 2Department of Psychology, Alma Mater Studiorum University of Bologna, BO 40126 Bologna, Italy

**Keywords:** workaholism, intensive smartphone use, sleep-wake cycle

## Abstract

Recent contributions have reported sleep disorders as one of the health impairment outcomes of workaholism. A possible factor affecting the sleep-wake cycle might be the intensive use of smartphones. The current study aimed to explore the role of intensive smartphone use in the relationship between workaholism and the sleep-wake cycle. Two serial multiple mediation models were tested on a sample of 418 employees, who filled self-report questionnaires measuring workaholism, use of smartphones, sleep quality and daytime sleepiness, using conditional process analysis for testing direct and indirect effects. Results supported our hypotheses regarding two serial multiple mediation models—that intensive smartphone use and poor sleep quality mediated the relationship between workaholism and daytime sleepiness, and that smartphone use and daytime sleepiness mediated the relationship between workaholism and poor quality of sleep. Although the use of a cross-sectional design and the snowball technique for collecting data can be considered as possible limitations, the current study is one of the first to document the potential detrimental role of the intensive smartphone use on the workaholism-sleep disorders relationship.

## 1. Introduction

In recent years there has been increasing attention towards the phenomenon of workaholism, a dysfunctional form of heavy work investment characterized by a set of recurrent behaviors (e.g., working for long hours) and cognitions (e.g., being mentally focused on work activities even when not at work) that have potentially strong negative implications for individual and organizational well-being and vitality [1,2,3]. Workaholism was firstly defined as an irresistible or uncontrollable need to work incessantly [4]. Schaufeli et al. [5] proposed that workaholism is characterized by two elements: Working excessively (exceptional amount of time and energy that workaholics devote to the work activity) and work compulsively (a strong and irresistible inner drive to work). Recently, some studies have shown a moderate/strong association between these two components [6]. However, it has been hypothesized that these two components are complementary and not distinct dimensions and, therefore, workaholism is often considered as a unitary phenomenon [7,8]. Another recent contribution related to workaholism an addiction perspective defining it as being overly concerned about work, being driven by an uncontrollable work motivation, and spending so much energy and effort on work that it impairs private relationships, spare-time activities and/or health [9]. Although it is difficult to estimate the prevalence of workaholism, some studies are cause of concern: Andreassen et al. [1] found a prevalence of 8.3% in a representative sample of Norwegian adult workers; Sussman et al. [10] found a prevalence of 10% in the American adult population.

A recent meta-analysis by Clark et al. [11] reported that workaholism was related to burnout, job stress, work-life conflict, and decreased physical and mental health. Recent contributions have also reported sleep disorders as one of the health impairment outcomes of being workaholic [12,13,14], although the methods in which sleep quality was conceptualized and measured were different. Some studies [12,15] focused their attention on investigating the relationship between workaholism and insomnia, a disorder that affects a significant portion of the world population, reporting a significant and positive association between the two constructs. Other studies have investigated sleep disorders more broadly. For instance, Caesens et al. [16] focused on the difficulty of falling asleep, waking up during the night and waking up tired. Kubota et al. [13,17] investigated the quality of sleep in general, sleep duration, sleep latency, the use of drugs that help sleep, sleep efficiency and problems during the day. In all these contributions, sleep disorders were often observed using measures of sleep quality at night, and considering the sleep consequences during the day separately—that is, daytime sleepiness. However, it is important not to separate the sleep-wake cycle, considering that the quality of a night’s sleep impacts the subsequent day and the level of daytime sleepiness impacts the following sleep night. Moreover, all these studies outlined a relationship between workaholism and sleep disorders, whereas little is known on the mechanisms underlying this relationship.

A recent contribution by Spagnoli et al. [14] reported evidence of the mediating role that job-related negative effect might play in the relationship between workaholism and anxiety before sleep. Translating it in practice, this means that job-related negative effect, fueled by workaholic tendencies, may be one of the reasons for which workaholics experience poor quality of sleep. However, other behavioral variables might play a mediating role in this process, such as, for example, the intensive use of mobile devices for working, such as the smartphone.

### The Mediating Role of Intensive Smartphone Use in the Workaholism–Sleep/Wake Cycle Relationship

We focused on smartphone because its intensive use has pervaded our lives. Smartphone is a wireless device with several functions, such as managing the calendar, making phone calls, browsing the Internet, and receiving and answering e-mails anytime, anywhere. The main reason for having a smartphone is to send and receive e-mails [18]. Thus, smartphone represents an important tool for working 24/7 and, consequently, we believe that workaholics—who are characterized by an inner drive that presses them to work in excess—tend to use it very intensively. Accordingly, Derks and Bakker [19] suggested that the smartphone, with its 24/7 availability, disturbs the important process of disengaging from work and to allow recovery. Some studies showed that the occurrence of a problematic mobile phone use seems to share some features with certain addictions, often referred to as behavioral addictions [20,21]. Since workaholism was defined as a behavioral addiction [22], we believe that workaholics intensively use their smartphone as a way to continuously being in touch with their work, even when they should be detached. Accordingly, Andreassen [23] claimed that workaholism is seen as a rather stable personal behavioral tendency—a tendency that most certainly can be exacerbated by the opportunities found in new technological innovations (laptops, smartphones, the Internet).

Following the above considerations, we aimed to shed light on the mechanism linking workaholism to the sleep-wake cycle by hypothesizing a possible role played by the intensive use of smartphones. Theoretically, the effect played by workaholism on the sleep–wake cycle via intensive smartphone use can be explained in light of the effort-recovery model [24], according to which the work effort can turn into load or strain reactions, including impaired sleep–wake cycle, if the worker experiences chronically high workload and incomplete recovery. Under incomplete recovery, the worker remains in a suboptimal energetic state, feeling tired of the previous work period and spending compensatory effort to perform adequately at work.

Since workaholics tend to work in excess and likely intensively use their smartphone to be connected to their work, their recovery process would be hampered, and this may manifest in worse sleep–wake cycle outcomes, such as poor sleep quality and daytime sleepiness.

Nevertheless, to the best of our knowledge, specific studies on the relationship between workaholism and the intensive use of smartphones are missing. Recent research reported negative health and psychological outcomes of intensive use of this communication device [25,26]. Specifically, sleep quality seems to be particularly compromised by intensive smartphone use [25,26]. Lanaj et al. [27] reported that prolonged use of a smartphone could compete with and disturb sleep, resulting in a trend toward insufficient sleep and degradation in sleep quality. A survey by National Sleep Foundation has found that 90% of Americans use light emitting (LE)-devices within an hour of their bedtime and that a greater use was associated with worse sleep outcomes at all ages [28]. Recently Chinoy et al. [29] also showed that unrestricted evening use of light-emitting tablet computers delayed self-selected bedtime and disrupted circadian timing and alertness. These results demonstrate that the use of such devices, by suppressing melatonin secretion, phase-delays the circadian clock, and disrupt the sleep–wake cycle, producing sleepiness during the morning hours.

Furthermore, it should be considered that the quality of daytime activities and experiences can impact on night sleep [30,31] and that a good quality of night sleep is a significant factor to allow an adequate daytime functioning [32]. For instance, it is possible to assume that workaholics are more prone to ruminate and/or to think about work after work-hours and even during sleep, considering that rumination affects sleep [33]. At the same time, it is possible to advance the idea that workaholics have low physical and mental levels, due to daytime sleepiness and try to counteract this sleepiness taking psychostimulant drugs (e.g., caffeine or tobacco), which impact the quality of sleep [34], or alternatively they think and work until late hours delaying (or losing) the optimal time to go to sleep, determining an insufficient sleep [35]. Since a stressful day determines a decrease of sleep quality (by increasing awakening during sleep) and a decrease in sleep quality determines an increase in daytime sleepiness [30], we adopted a comprehensive approach to assess the potential implications of smartphone use on the sleep–wake cycle, studying both the characteristics of night sleep and daytime functioning. Considering the negative effects of LE electronic devices on sleep, it is expected that intensive use of smartphones, such as the one characterizing workaholic subjects, can significantly impair sleep quality, also affecting the subsequent daytime functioning [36]. Accordingly, we hypothesized two serial multiple mediation models, depicted in Figure 1. In model 1 we expected that intensive smartphone use and poor sleep quality at night mediate the relationship between workaholism and daytime sleepiness, whereas in Model 2 we hypothesized that intensive smartphone use and daytime sleepiness mediate the relationship between workaholism and poor sleep quality at night. Therefore, we put forward the following two hypotheses:
**H1:** Intensive smartphone use and poor sleep quality mediate the relationship between workaholism and daytime sleepiness.

In particular, we expected a significant and positive relationship between workaholism, intensive smartphone use, poor sleep quality and daytime sleepiness.
**H2:** Intensive smartphone use and daytime sleepiness mediate the relationship between workaholism and poor sleep quality.

Specifically, we expected: a significant and positive relationship between workaholism and intensive smartphone use; a significant and positive relationship between intensive smartphone use and daytime sleepiness; and a significant and positive relationship between daytime sleepiness and poor sleep quality. In testing the proposed paths, we controlled for some meaningful variables, namely workload, gender, age, ad type of the job to better weight the unique role of workaholism in the model. Actually, a recent meta-analysis showed that workload constitutes one of the most predictive work domain variable of workaholism [11], whereas despite gender and age were not found to be correlated to workaholism from the meta-analysis, recent studies underpinned the need to consider these variables in the study of workaholism [1,14]. Regarding the job type, some authors pointed out that workers performing particular jobs, such as, for example, freelancers, could be more exposed to the risk of workaholism [37]. Since in our sample, different jobs are comprised, we believed it was important to include it as a control variable.

## 2. Materials and Methods

### 2.1. Participants

A total of 418 Italian workers, 61.7% of women, were involved in the study. Age ranged between 19 and 72 years (Mean = 44.01, S.D. = 12.56). They worked as teachers (42.5%), clerk (16.6%), freelancers (20.4%), managers (15.1%), and entrepreneurs (5.3%).

### 2.2. Procedure

Participants were administered a paper and pencil self-report questionnaire. The data collection was carried out by graduating students of work and organizational psychology courses as part of their Master’s degree thesis. Students were first trained on how to present the study and its objectives to potential participants, including how to take advantage of the snowball sampling technique. Subsequently, students were asked to identify acquaintances in their social network and to propose them to take part in a study on work-related health and well-being.

### 2.3. Measures

#### 2.3.1. Workaholism

Workaholism was measured by using the 10-item version of the Dutch Work Addiction Scale (DUWAS) adapted in Italian by Balducci et al. [38]. The DUWAS investigates the respondent’s feelings about his/her work, which reflect the two components of workaholism (i.e., working compulsively, WC, and working excessively, WE). Similar to previous studies [39], the two workaholism components were collapsed in one workaholism dimension for deriving an overall workaholism score. Example items are the following: “*I feel that there’s something inside me that drives me to work hard*” (WC) and “*I stay busy and keep many irons in the fire*” (WE). Responses were given on a five-point scale varying from 1 (“Never or almost never”) to 5 (“Almost always or always”).

#### 2.3.2. Intensive Smartphone Use

Intensive smartphone use was measured with two items proposed by Derks and Bakker [19] plus an additional ad hoc item. Items were formulated assuming intensive smartphone use as a habit, that is, as a trait-like phenomenon. The three items were: “*When my smartphone blinks to indicate new messages, I cannot resist checking them*”; “*I use my smartphone intensively*”; “*I am usually online until sleep time*”. All items were rated on a five-point Likert scale from 1 (totally disagree) to 5 (totally agree).

#### 2.3.3. Sleep-Wake Cycle

The Mini-Sleep Questionnaire [40] was used to measure sleep quality and daytime sleepiness. It consists of 10-items, each with a five-point Likert scale. Participants had to indicate the frequency (from never to always) of occurrence of sleep-wake cycle circumstances in the last seven days. Six items are related to sleep quality (e.g., item 1: “*Difficulty falling asleep*”), while four items are related to excessive daytime sleepiness (e.g., item 9: “*Excessive daytime sleepiness*”).

#### 2.3.4. Workload

The workload was measured by three items (e.g., “*I have to work very fast*”) from the Job Content Questionnaire [41]. Responses were given on a five-point scale varying from 1 (totally disagree) to 5 (totally agree).

For the purpose of the analyses, the score on each investigated construct/phenomenon was derived by computing the respective scale mean.

### 2.4. Statistical Analysis

Cronbach’s alphas and zero-order correlations (Pearson’s r) were used to assess the scales’ internal consistencies and examine associations between variables. The hypotheses concerning direct and indirect effects were tested through bootstrapping [42], a nonparametric resampling procedure that does not assume normality and involves the extraction of several thousand subsamples (5000, in our case) from a dataset. Through bootstrapping, the distribution of effects is empirically approximated and used for calculating confidence intervals [43]. Using conditional process analysis in SPSS 22 (IBM, Armonk, NY, USA), we calculated two multiple mediation models (see Figure 1) corresponding to the conceptual model number 6 of Hayes [42] templates. Specifically, the two models were: Model 1, which tested the mediating role of intensive smartphone use and poor sleep quality in the relationship between workaholism and daytime sleepiness, and Model 2, which tested the mediating role of intensive smartphone use and daytime sleepiness in the relationship between workaholism and poor sleep quality. The difference between the two models only concerned the order of the sleep-wake cycle variables. A different order of sleep–wake variables was tested in these two models because it allowed to analyze the relationship between workaholism and sleep-wake cycle, which is regulated by the two-process model during the day with a homeostatic process, representing sleep debt during wakefulness, and a circadian pacemaker, synchronizing periodically day and night [44].

Thus, it was necessary to test both the models for a twofold reason, theoretical and empirical. As it is not conceivable to establish a right and the fixed order of the two components of the sleep-wake cycle (daytime sleepiness stems from low sleep quality and vice versa) we had to consider both the two paths for empirically testing the two possible model combinations. Most probably, in the first model, the workaholic trait influenced the intensive use of smartphones, which affected poor sleep quality and, in turn, subsequent daytime sleepiness. A similar idea underpinned the second model as well, with the only exception that the relationship between workaholism and poor sleep quality was mediated by the intensive use of smartphones and daytime sleepiness, given that for the former variable workaholics can delay the optimal time to go to sleep, due to LE-device, as well as remaining awake to work, and for the latter variable workaholics can adopt maladaptive behavior to counteract sleepiness.

### 2.5. Ethics

The procedure was in accordance with the standards of the national law of data treatment, which is strictly followed by the University of Campania “Luigi Vanvitelli” and University of Bologna Alma Mater Studiorum (Italy). Since there was no medical treatment or other procedures that could cause psychological or social discomfort to participants, who were all healthy adult subjects anonymously involved, additional ethical approval was not required. The research was conducted in line with the Helsinki Declaration [45], as well as the data protection regulation of Italy (Legislative Decree No. 196/2003). Participation in the study was voluntary and not rewarded; data collection and analysis were anonymous. A cover letter attached to the questionnaire provided information about the study aims, guarantees about anonymity, voluntary participation and data treatment, and instructions for filling out the questionnaire. When agreeing to fill out the questionnaire, all study participants provided their informed consent.

## 3. Results

Table 1 shows the zero-order correlations between the study variables and their reliability measured by Cronbach’s alpha coefficient. All the reliability coefficients resulted above, or nearly above in the case of workload, the commonly-accepted threshold of 0.70. Workaholism positively correlated with intensive smartphone use, poor sleep quality, daytime sleepiness and workload. Intensive smartphone use positively correlated with poor sleep quality, daytime sleepiness and job type, whereas it was negatively associated with age. Gender was only positively related to daytime sleepiness.

Additionally, according to Andreassen et al. [1], we computed prevalence of workaholism in the current sample. Results reported that 12.67% of the sample showed a workaholism’s score between 4 to 5. Moreover, quartiles’ mean analysis showed a workaholism score equal to and greater than 3.6 for participants at the 75° percentile or above (*N* = 123). In sum, these results pointed out a relevant incidence of potential workaholics in the current sample.

Table 2 and Table 3 report the results of the two models tested. In Model 1, evidence of two indirect relationships were found. In particular, the mediating role of poor sleep quality in the relationship between workaholism and daytime sleepiness (indirect effect B = 0.20, LLCI = 0.12 ULCI = 0.30), and the serial multiple mediating role of intensive smartphone use and poor sleep quality in the relationship between workaholism and daytime sleepiness (indirect effect B = 0.03, LLCI = 0.01 ULCI = 0.06). The total ‘effect’ of the mediating model tested was 0.40 (LLCI = 0.00, ULCI = 0.27). These results supported our first hypothesis (H1).

Moreover, we tested Model 2, in which we merely switched the order of the sleep–wake cycle variables. We found two significant indirect relationships: the mediating effect of daytime sleepiness in the relationship between workaholism and poor sleep quality (indirect effect B = 0.22, LLCI = 0.13 ULCI = 0.31), and the serial mediating effect of intensive smartphone use and daytime sleepiness in the relationship between workaholism and poor sleep quality (indirect effect B = 0.02, LLCI = 0.01 ULCI = 0.05). Thus, H2 was also supported.

While we found both the serial multiple mediation models to be significant, following Hayes [42], we conducted a comparison of the fit of the two models considering the R^2^ coefficient. This showed that Model 1 explained a very slight part of the variance of the independent variable (R^2^ = 0.47), which for this model is daytime sleepiness, more than the variance of the independent variable (R^2^ = 0.46) explained in Model 2, which is poor sleep quality. This difference is insignificant and, thus, we can conclude that both the models have a satisfactory fit and explain approximately the same amount of variance in the respective dependent variable.

## 4. Discussion

The current study explored the mediating role of intensive smartphone use in the mechanism linking workaholism to the sleep-wake cycle. Several studies have addressed the negative health impact, and also specifically sleep disorders, of intensive or problematic use of smartphones in adolescent samples [26], whereas very few studies focused the same phenomenon on adults samples [25]. Moreover, to the best of our knowledge, the current study is one of the first to address this phenomenon (i.e., the relationship between smartphone use and sleep-wake cycle) in association with workaholism.

Drawing on the effort-recovery model [24] and adopting an interdisciplinary perspective encompassing psycho-physiological research contributions we hypothesized that the recovery process in workaholics could be limited and incomplete, due to the continuous and intensive use of the smartphone, leading to consequences on the sleep-wake cycle. We adopted a comprehensive approach to the study of the sleep-wake cycle by using measures of sleep quality and of daytime sleepiness.

Our expectations concerning the hypothesized models were confirmed: On the one hand, both intensive smartphone use and poor sleep quality could have an important role in the way workaholics experience daytime sleepiness, while on the other hand intensive smartphone use and daytime sleepiness mediated the relationship between workaholism and poor sleep quality independently and separately. Intensive smartphone use was directly and significantly connected to both poor sleep quality and daytime sleepiness. The result is consistent with Chinoy et al. [29], who showed that evening use of light-emitting devices impair next-morning alertness, and it is possible to hypothesize that a workaholic individual continues to use a smartphone in the evening hours after usual daytime working hours. This is also consistent with previous research showing that the negative effects of light-emitting diodes (i.e., blue light) on poor sleep quality and quantity are found when exposure to light-emitting diodes is administered in the evening - that is, when the circadian timing system is most vulnerable to light [46,47].

Furthermore, we found that the relationship between workaholism and poor sleep quality was also mediated by daytime sleepiness in model 2. This result is consistent with previous results on the relationship between workaholism and sleep problems during the day [13,17]. Even if we did not control for a time-of-day effect on smartphone use, our results clearly showed that availability of LE-devices not only disturb sleep quality throughout disruption of circadian rhythmicity, but can also induce activation of the neurophysiological systems which sustain wakefulness and contrast the sleepiness generated by the homeostatic sleep drive [48,49,50], further disturbing natural sleep–wake processes.

This explanation is valid also considering that workaholics remain linked to their job longer through their smartphones, thus increasing their exposure to a factor that disturbs sleep, such as delaying (or losing) the optimal time to go to sleep, determining an insufficient (restorative) sleep night.

Although our study had the merit to shed some light on a possible mechanism through which workaholism is related to sleep disorders, some important limitations need to be taken into account. First, the cross-sectional nature of the study does not allow establishing the causal order between the focused variables. Thus, although the mediating model at the core of the study stems from a sound theoretical basis, the results should be interpreted with caution. However, the evidence found will provide useful insights for supporting the possibility of further studies to confirm the mediating model longitudinally or through a diary study. Future studies could, for example, assess workaholism with a preliminary survey and then intensive smartphone use, daytime sleepiness and poor sleep quality with repeated daily measures fitting the postulated causal ordering, thus, providing more strong evidence on cause-effect relationships between the variables.

Second, since self-report measures were adopted, the results might be influenced by the participants’ acquiescence, need for social desirability, and the emerged parameter estimates may have been contaminated by common method bias [51]. Future studies should adopt multisource and objective measures. For example, as far as sleep disorders are concerned, actigraphy could be used for detecting more objective measures of quality and quantity of sleep. Third, the snowball technique for data collection contravenes many of the assumptions supporting conventional notions of random selection and representativeness. However, it is not uncommon to use such a sampling strategy in organizational research [52].

Fourth, although we controlled the tested model for workload, gender, age, and type of job other individual variables highly correlated to workaholism, such as neuroticism and perfectionism, should have been included in order to better weight the impact of workaholism in the model.

Finally, the present study is based on a definition of workaholism according to which its main components are working excessively and working compulsively. However, multiple definitions of workaholism can be found in the literature [11] and, as a consequence, different measurement scales. Thus, future research in this area would also benefit from theoretical work aimed at reconciling the different perspectives available on workaholism, so to come to a more shared definition and measurement of the phenomenon.

There are several interesting practical implications stemming from our study. Mobile phones are becoming more and more equipped, and the attraction to use them is likely to increase. Hence, the individual’s capacity to control its use is important. As Bert et al. [25] claimed, at the societal level, public health prevention strategies could foster optimum sleep habits, and information regarding the effects of intensive electronic devices use on the sleep status. Considering the results of this research, it can be recommended for health professionals to ask questions to patients about the sleep–wake cycle, considering that a good sleep-wake pattern influences social functioning and the adoption of healthy behaviors. Health professionals should also ask patients questions about the causes of a possible inadequate sleep-wake pattern, including workaholism, stressing the negative impact of excessive work and intensive use of electronic devices on the quality of individual life. Finally, health professionals should determine when referral to a sleep specialist is needed. In fact, sleep disorders should be quickly diagnosed because their consequences could also have important clinical and socioeconomic issues: Increased hospitalizations, absenteeism at work and a higher risk of road accidents. According to these implications, for example, both models could show the relevance to treating sleep-wake problems among adults, and specifically among workaholics, given that sleep disorders compromise the sleep-wake cycle, affecting sleep (hyposomnia) and/or wake (hypersomnia). For instance, a great number of insomnia patients take hypnotic drugs to sleep and large amounts of psychostimulant drugs, like caffeine, have demonstrated negative effects, such as increased level of anxiety and sleep impairment [53]. Hygiene habits of sleep, attitudes, beliefs and mental arousal play a pivotal role in the onset and maintenance of sleep-wake problems [54]. In a 24/7 society with the possibility to work 24/7 with a smartphone, there is the tendency to reduce sleep hours to increase the waking hours and, consequently, the time available for work, especially for workaholics. However, it is important to be aware of the cyclical nature of human physiology, and any attempt to counteract it can have relapses at different levels, starting from the well-being of the person.

## 5. Conclusions

The current study is one of the first to reveal the potentially detrimental effect of intensive smartphone use on workaholism outcomes, namely sleep disorders. The availability of the smartphone for working is certainly an important resource in the modern concept of working anywhere and anytime. However, it should be kept in mind that its intensive use, especially in workaholic workers, should be reduced in order to avoid its possible disrupting consequences, such as, for example, disorders connected to the sleep-wake cycle. Moreover, by adopting an interdisciplinary perspective, the current study proposes a more comprehensive approach to the study of this workaholism’ outcome by considering the sleep-wake cycle and work-related habits, such as the intensive smartphone use.

## Figures and Tables

**Figure 1 ijerph-16-03517-f001:**
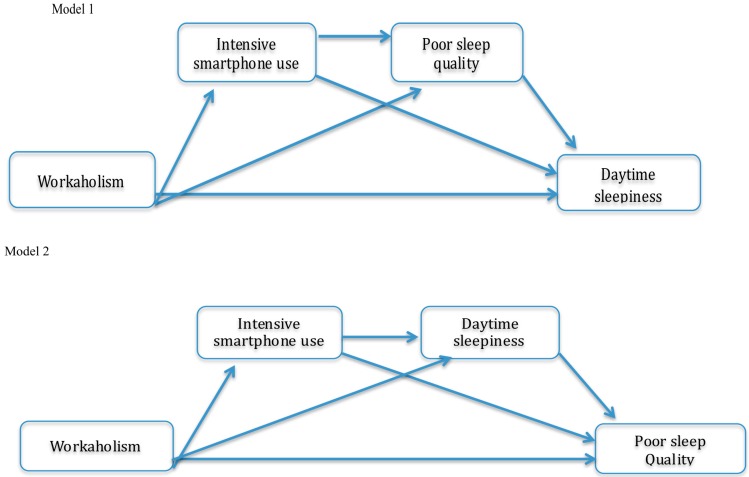
The two hypothesized serial multiple mediation models.

**Table 1 ijerph-16-03517-t001:** Descriptives, reliabilities (Cronbach’s Alphas in italic in diagonal) and inter-correlations of the focused variables.

Variables	M	S.D.	1	2	3	4	5	6	7
1. Workaholism	3.19	0.36	*0.77*						
2. Intensive smartphone use	3	1.07	0.24 **	*0.76*					
3. Poor sleep quality	2.43	0.77	0.23 **	0.18 **	*0.77*				
4. Daytime sleepiness	2.61	0.83	0.29 **	0.19 **	0.65 **	*0.77*			
5. Gender #	-	-	0.05	−0.05	0.07	0.12 **	-		
6. Age	44.01	12.56	−0.09	−0.20 *	0.10	−0.10	0.04	-	
7. Workload	3.6	0.70	0.40 **	0.08	0.08	−0.01	0.02	−0.16	*0.66*
8. Job type	-	-	0.03	0.14 *	−0.06	0.02	−0.30 **	−0.27 **	0.11 *

# = Gender was coded as 1 = male, 2 = female; * = *p* Value <0.05; ** = *p* Value < 0.001.

**Table 2 ijerph-16-03517-t002:** Model 1—Results of the direct and indirect effects.

Models	B	LLCI	ULCI	R^2^
**Model 1 a: Mediation of smartphone use in the relationship between workaholism and poor sleep quality**				
Outcome variable: Smartphone use				0.09 *
Workaholism	0.41 **	0.24	0.58	
Covariate: Job type	0.06	−0.01	0.13	
Covariate: Gender	−0.03	−0.24	0.18	
Covariate: Age	−0.01	−0.02	−0.01	
Covariate: Workload	−0.06	−0.22	0.09	
**Model 1 b: Mediation of smartphone use in the relationship between workaholism and poor sleep quality**				
Outcome variable: Poor sleep quality				0.10 **
Workaholism	0.30 **	0.18	0.43	
Intensive smartphone use	0.12 *	0.05	0.18	
Covariate: Job type	−0.02	−0.70	0.03	
Covariate: Gender	0.25 *	0.06	0.44	
Covariate: Age	0.01	−0.002	0.01	
Covariate: Workload	−0.001	−0.14	0.14	
**Model 1 c: Mediation of smartphone use in the relationship between workaholism and daytime sleepiness**				
Outcome variable: Daytime sleepiness				0.47 **
Workaholism	0.41	0.27	0.54	
Intensive smartphone use	0.02	−0.04	0.07	
Poor sleep quality	0.67 **	0.59	0.75	
Covariate: Job type	0.02	−0.01	0.06	
Covariate: Gender	0.17	0.04	0.29	
Covariate: Age	−0.01	−0.01	−0.01	
Covariate: Workload	−0.01	−0.09	0.09	
**Indirect effects**				
Workaholism-intensive smartphone use-daytime sleepiness	−0.01	−0.01	0.03	
Workaholism-poor sleep quality-daytime sleepiness	0.2	0.12	0.3	
Workaholism-intensive smartphone use-poor sleep quality-daytime sleepiness	0.03	0.01	0.06	
**Total effect**	0.4	0	0.27	

Note: B = unstandardized estimated; LLCI = Lower Level Confidence Interval; ULCI = Upper Level Confidence Interval.

**Table 3 ijerph-16-03517-t003:** Model 2—Results of the direct and indirect effects.

Models	B	LLCI	ULCI	R^2^
**Model 2 a: Mediation of smartphone use in the relationship between workaholism and daytime sleepiness**				
Outcome variable: Smartphone use				0.09 *
Workaholism	0.41 **	0.24	0.58	
Covariate: Job type	0.06	−0.01	0.14	
Covariate: Gender	0.03	−0.25	0.18	
Covariate: Age	−0.01	−0.02	0.01	
Covariate: Workload	−0.06	−0.22	0.08	
**Model 2 b: Mediation of smartphone use in the relationship between workaholism and daytime sleepiness**				
Outcome variable: Daytime sleepiness				0.12 **
Workaholism	0.36 **	0.23	0.5	
Intensive smartphone use	0.10 *	0.02	0.37	
Covariate: Job type	0.01	−0.04	0.06	
Covariate: Gender	0.21 *	0.04	0.37	
Covariate: Age	−0.01	−0.01	0.01	
Covariate: Workload	−0.08	−0.20	0.03	
**Model 2 c: Mediation of smartphone use in the relationship between workaholism and poor sleep quality**				
Outcome variable: Poor sleep quality				0.46 **
Workaholism	0.09	−0.01	0.18	
Intensive smartphone use	0.06	0.01	0.11	
Daytime sleepiness	0.59 **	0.52	0.66	
Covariate: Job type	−0.02	−0.06	0.01	
Covariate: Gender	−0.06	−0.18	0.05	
Covariate: Age	0.01	0.01	0.01	
Covariate: Workload	−0.06	−0.14	0.02	
**Indirect effects**				
Workaholism-intensive smartphone use-poor sleep quality	0.01	−0.01	0.05	
Workaholism-daytime sleepiness-poor sleep quality	0.22	0.13	0.31	
Workaholism-intensive smartphone use-daytime sleepiness-poor sleep quality	0.02	0.01	0.05	
**Total effect**	0.34	0.22	0.46	

Note: B = unstandardized estimated; LLCI = Lower Level Confidence Interval; ULCI = Upper Level Confidence Interval.

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
