# Peer review of "Workaholism, Intensive Smartphone Use, and the Sleep-Wake Cycle: A Multiple Mediation Analysis"

_ijerph, 2019, doi:10.3390/ijerph16193517_

Round 1

Reviewer 1 Report

In this manuscript, the authors explored the association of the workaholism with sleep quality. They used the serial multiple mediation models to consider the mediation effect of intensive smartphone use.  The hypothesized models and their analysis are well described in general. Major concerns are as follows: 

Major points: 

The authors provided two models by changing the order of the sleep-wake cycle variables. Which model is better in terms of goodness of fit?  With only one variable among Smartphone use and sleep-wake cycle variables, the single mediation analysis results would be interesting for comparison.   

Minor points: 

There are some typos (even in the Title, "Workaohlism"). Page6, Line221: the numbers in the sentence about "total effect" is not the same as those in Table 2. 

Author Response

Reviewer 1 Comments and Suggestions for Authors

In this manuscript, the authors explored the association of the workaholism with sleep quality. They used the serial multiple mediation models to consider the mediation effect of intensive smartphone use.  The hypothesized models and their analysis are well described in general. Major concerns are as follows: 

Major points: 

The authors provided two models by changing the order of the sleep-wake cycle variables. Which model is better in terms of goodness of fit?  With only one variable among Smartphone use and sleep-wake cycle variables, the single mediation analysis results would be interesting for comparison.   

Thank you very much for this interesting point. According to your question and suggestion we added a brief part in the results section addressing the comparison of fit between the two models. Following Hayes (2017) we used the R2 coefficient for such a comparison: “R2 can be thought of as the fit of a model in which Y is estimated to be Y for every case in the data, as if one entirely ignored the information contained in the antecedents that could be used to estimate Y with greater precision than the use of this naive strategy of predicting Y for every case… R2 quantifies the distance the best fitting linear regression model has travelled between this naive reference model and a perfectly fitting model.” (pag. 55; Hayes, 2017).

This is the part that we added in the text : “While we found both the serial multiple mediation models to be significant, following Hayes (2017) we conducted a comparison of fit of the two models considering the R2 coefficient. This showed that Model 1 explained a very slight part of variance of the independent variable (R2=.47), which for this model is daytime sleepiness, more than the variance of the independent variable (R2=.46) explained in Model 2, which is poor sleep quality. This difference is insignificant and, thus, we can conclude that both the models have a satisfactory fit and explain approximately the same amount of variance in the respective dependent variable”.

Regarding your suggestion to compare the single mediations, results are already available in Table 2. In fact Model 1b (intensive smartphone use mediates the relationship between workaholism and poor sleep quality) and Model 2b (intensive smartphone use mediated the relationship between daytime sleepiness) are related to the single mediations tested in the multiple mediation model. As you can note, the R2 for the two model is rather similar (Model 1b – R2=.10 and Model 2b – R2=.12).

Minor points: 

There are some typos (even in the Title, "Workaohlism"). Page6, Line221: the numbers in the sentence about "total effect" is not the same as those in Table 2. 

Thank you! We went through the manuscript for checking the typos and also corrected the numbers for total effect in Table 2 that by mistake were erroneous.

Reviewer 2 Report

This study covers an interesting and relevant topic nowadays by proposing underlying mechanisms linking workaholism and sleep disturbance. 

Even though I do not deny the relevance of this study, I consider that introduction is underdeveloped. For example, is not clear the theoretical framework that the study is based on. Additionally, the concept of workaholism should also be more developed, regarding its definition and consequences, as well as additional linking mechanisms that past research proposes and, consequently, the importance of suggesting additional mechanisms above and beyond the already proposed ones. Thus, taking these arguments into account, the contribution of the study should also be more clearly stated.

For me is not completely clear why it is relevant to propose two alternative models that include the same variables in a different order. 

Regarding discussion, the theoretical and practical implications of the study are also two summarized, I consider that you should describe more deeply the implications of the study (referring the two models separately). 

I also consider that you should focus more on future research avenues besides those that comes from the limitations of the study. Once again, your conclusion is also two summarized, try to developed it further.

I suggest you take another pass through the manuscript to clean up minor grammar and usage issues (e.g. the word workaholism in the title is misspelled).

Author Response

Reviewer 2 Comments and Suggestions for Authors

This study covers an interesting and relevant topic nowadays by proposing underlying mechanisms linking workaholism and sleep disturbance. 

Even though I do not deny the relevance of this study, I consider that introduction is underdeveloped. For example, is not clear the theoretical framework that the study is based on. Additionally, the concept of workaholism should also be more developed, regarding its definition and consequences, as well as additional linking mechanisms that past research proposes and, consequently, the importance of suggesting additional mechanisms above and beyond the already proposed ones. Thus, taking these arguments into account, the contribution of the study should also be more clearly stated.

Thank you for this very helpful comment. We revised the manuscript by elaborating a bit more on the concept of workaholism and also by briefly mentioning the theoretical framework of the effort-recovery model (Meijman & Mulder, 1998), which fits very well as a background for the hypotheses we tested. We also slightly extended the discussion and the conclusion section by arguing a little bit more about our purposes, the results obtained and the practical implications for the individual and society. We hope that the current version of the manuscript can make our contribution clearer.

For me is not completely clear why it is relevant to propose two alternative models that include the same variables in a different order. 

Thank you also for this point. We tried to clarify a bit better why it could be interesting and useful, in the context of the present study, to test for the two models. Specifically, in the statistical analysis section we added the following sentence: “Thus, it was necessary to test both the models for a twofold reason, theoretical and empirical. As it is not conceivable to establish a right and fixed order for the two components of the sleep-wake cycle (daytime sleepiness stems from low sleep quality or vice versa) we considered both the paths and empirically tested the two alternative models”. Moreover, we added some more clarification in the whole text. We hope that this is now clearer.

Regarding discussion, the theoretical and practical implications of the study are also two summarized, I consider that you should describe more deeply the implications of the study (referring the two models separately). 

I also consider that you should focus more on future research avenues besides those that comes from the limitations of the study. Once again, your conclusion is also two summarized, try to developed it further.

Thank you for raising this important point. We elaborated a bit more the discussion. Also, in the limitation section we stressed a little bit more the need to conduct longitudinal and diary studies using a more robust approach for testing the theoretical model. Also, we briefly outlined the issue of the controversy on the definition of workaholism and the consequent fragmented literature.

I suggest you take another pass through the manuscript to clean up minor grammar and usage issues (e.g. the word workaholism in the title is misspelled).

Thank you! We went through the manuscript for checking the typos and also corrected the numbers for total effect in Table 2 that by mistake were erroneous.

Reviewer 3 Report

It is helpful to state which software was used for analysis.

Author Response

Reviewer 3 Comments and Suggestions for Authors

It is helpful to state which software was used for analysis.

Thank you! We included the software used in the statistical analysis section.